

# Real-time DDoS flood attack monitoring and detection (RT-AMD) model for cloud computing

Omaimah Bamasag[1], Alaa Alsaeedi[2], Asmaa Munshi[3], Daniyal Alghazzawi[4], Suhair Alshehri[5] and Arwa Jamjoom[4]

[1] Department of Computer Science, Faculty of Computing and Information Technology, King Abdulaziz University, Jeddah, Saudi Arabia
[2] Department of Computer Science, University of Jeddah, Jeddah, Saudi Arabia
[3] Cybersecurity Department, University of Jeddah, Jeddah, Saudi Arabia
[4] Department of Information Systems, Faculty of Computing and Information Technology, King Abdulaziz University, Jeddah, Saudi Arabia
[5] Department of Information Technology, Faculty of Computing and Information Technology, King Abdulaziz University, Jeddah, Saudi Arabia

## ABSTRACT

In recent years, the advent of cloud computing has transformed the field of computing and information technology. It has been enabling customers to rent virtual resources and take advantage of various on-demand services with the lowest costs. Despite the advantages of cloud computing, it faces several threats; an example is a distributed denial of service (DDoS) attack, which is considered among the most serious. This article presents real-time monitoring and detection of DDoS attacks on the cloud using a machine learning approach. Naïve Bayes, K-nearest neighbor, decision tree, and random forest machine learning classifiers have been selected to build a predictive model named "Real-Time DDoS flood Attack Monitoring and Detection RT-AMD." The DDoS-2020 dataset was constructed with 70,020 records to evaluate RT-AMD's accuracy. The DDoS-2020 contains three protocols for network/transport-level, which are TCP, DNS, and ICMP. This article evaluates the proposed model by comparing its accuracy with related works. Our model has shown improvement in the results and reached real-time attack detection using incremental learning. The model achieved 99.38% accuracy for the random forest in real-time on the cloud environment and 99.39% on local testing. The RT-AMD was evaluated on the NSL-KDD dataset as well, in which it achieved 99.30% accuracy in real-time in a cloud environment.

# INTRODUCTION

The emergence of cloud computing has gained much attention due to its various features such as cost-effectiveness and on-demand service provision. Cloud computing is a shared environment (multi-tenancy) between more than one user, using the same physical resources. Despite its advantages, the shared environment concept may threaten the security and availability of provided services. A cloud services provider (CSP) must have the ability to ensure the security and availability of resources to maintain the commitment

Corresponding author
Alaa Alsaeedi,
aalsaeedi0034.stu@uj.edu.sa

to customers, called the service level agreement (SLA). Cloud computing is becoming more popular as more people and companies are attracted to employing it in their businesses. Its utilization is of high benefit; however, security remains a serious problem, especially in the public cloud environment.

This study will investigate current work in DDoS attacks targeting cloud services and propose an efficient model to detect DDOS flooding attacks at the network/transport-level. This model is called the Real-Time DDoS flood Attack Monitoring and Detection (RT-AMD) Model, which aims to enhance cloud services security by protecting all resources in a cloud environment from DDoS attacks. It is characterized by being real-time as it monitors the cloud environment and alerts any attempted attack in real-time. The administrator will be notified of this incident with a timely alert message. The notification message contains all the information on the attack to facilitate the administrator in dealing with it.

The contribution of this research is twofold: the first is to evaluate machine learning algorithms for the collected dataset; the second is to improve the performance to reach real-time attack detection.

## BACKGROUND

Recently, cloud computing has gained much attention as it has widespread impacts across different fields such as information technology, business, software engineering, and data storage. The cloud environment provides resources to customers in a virtual way with high efficiency and low cost. For example, it enables the users to experiment with software products before purchasing them and use storage capacity at a low cost compared to buying it in traditional ways. The National Institute of Standards and Technology (NIST) defines cloud computing as "a model for enabling convenient, resource pooling, ubiquitous, on-demand access which can be easily delivered with different types of service provider interaction" (*Zissis & Lekkas, 2012*).

A cloud environment is characterized by many features such as manageability, scalability, availability, security, on-demand service, expedience, ubiquity, multitenancy, elasticity, and stability. The services delivery in a cloud environment was categorized into three main models: infrastructure as a service (IaaS), platform as a services (PaaS), software as a service (SaaS), and everything as a service (XaaS) either in public, private, community, or hybrid cloud as defined by NIST (*Singh, Jeong & Park, 2016*). Figure 1 shows the services delivery model details with examples.

Many security challenges are faced by CSP, of which the main one is the trust that must be in place between CSP and the cloud customers. Trust is how the provider can protect the customer data from any breach (*Ghaffari, Gharaee & Arabsorkhi, 2019*). One of the popular features of the cloud environment is multi-tenancy and virtualization. Many customers share physical resources, which constitutes a considerable challenge in making such an environment secure (*Singh, Jeong & Park, 2016*).

Shared data in the cloud can create risks of customers' data being lost or used by an unauthorized third party. There are many other types of cyberattacks on cloud security allowed by system and application vulnerabilities, such as account hijacking, malicious

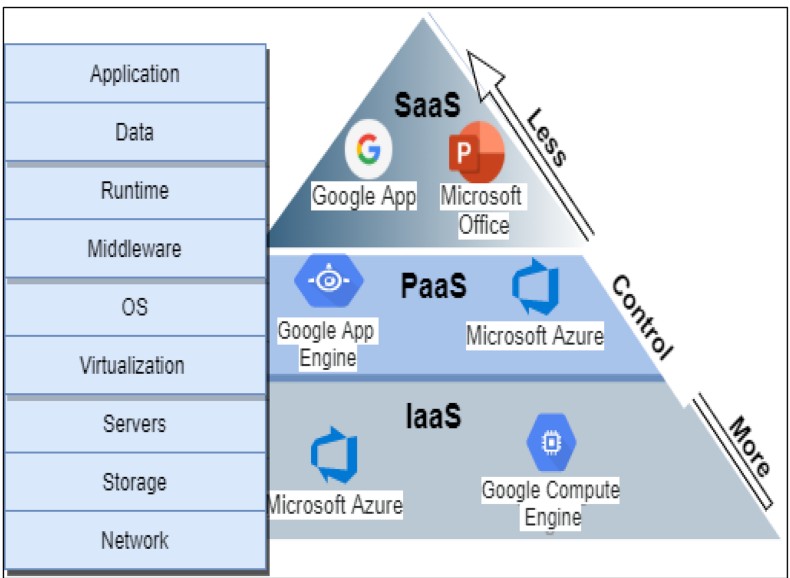

**Figure 1 Cloud services delivery models.**

insiders, data loss, denial, and distributed denial of service. These attacks have a substantial negative impact on confidentiality, integrity, and availability of data.

The availability of cloud services is one of the most critical CSP goals. Unavailability adversely affects CSP and cloud customers. DoS and DDoS attacks are the main threats leading to a cloud service's unavailability. DoS is a cyberattack where an attacker aims to make the systems and servers unavailable, preventing customers from accessing the servers and resources (*Douligeris & Mitrokotsa, 2004*). DoS attacks launched in a distributed manner to speed up the consumption of the resources for one or many targets are called distributed denial of service attacks (DDoS) (*Douligeris & Mitrokotsa, 2004*). DDoS attack types are explained in the following subsection.

## DDoS attack types

DDoS attacks are classified based on targeted protocols such as the network/transport or application level.

- Network/Transport level DDoS attacks: These attacks occur mostly when using network and transport layer protocols such as TCP, UDP, and ICMP. These attacks are further categorized into three types:

  1. Volume attacks: The attacker aims to consume all the resources of the target servers and make them unavailable by sending many packets (bandwidth/flooding attack) such as TCP flood, ICMP flood, *etc.*
  2. Protocol attacks: In this type, the attack consumes all resources and intermediate connection media as a firewall by exploiting protocol vulnerabilities and bugs as TCP SYN flood, TCP SYN-ACK flood, *etc.*

3. Reflection and amplification attacks: Attempts to consume the victim's resources by sending fake request messages (such as ping requests) through spoofing the victim's IP address to the reflectors. The reflectors send a high volume of response messages to the victim's IP address such as Smurf attacks.

- Application-level DDoS attacks: These attacks aim to consume services' resources or cause starvation of resources to disrupt customers through establishing requests, overloading the application servers. The most popular type of attack at this level is HTTP flooding attacks. Many studies have classified DDoS attack at the application level based on the following categories (*Jaafar, Abdullah & Ismail, 2019*):

1. Session flooding attack: Servers' resources are disabled from being launched when session request rates are high. These requests usually are higher than those generated by valid users.
2. Request flooding attack: Sends sessions that contain more requests than the valid users.
3. Asymmetric attack: Wastes resources such as CPU and memory of the server by sending sessions with high-workload requests.
4. Slow request/response attack: Uses all server resources by sending incomplete requests slowly to keep the servers in the waiting state to receive data.

## Intrusion detection system

An intrusion detection system (IDS) is a device or software tool that identifies unusual events by monitoring the network traffic to distinguish the normal from abnormal behaviors (*Kaur, Kumar & Bhandari, 2017*). IDS is classified into three main categories based on the analysis method. The choice between methods depends on several factors, such as the anomaly type, applied environment, security level required, and the cost (*Kaur, Kumar & Bhandari, 2017*).

The IDS methods classifications are signature-based, anomaly-based, and hybrid detection (*Alzahrani & Hong, 2017*). Signature-based detection, also called knowledge-based or rule-based detections, is suitable for detecting known attacks by comparing captured behavior. Anomaly-based detection, also known as behavior-based, is useful to detect unknown attacks. These techniques compare the observed behavior with normal behavior to detect abnormal events. Hybrid-based detection works by combining the detection techniques mentioned above. The performance of this detection depends on the types of techniques chosen. Table 1 shows the advantage and limitations of these detection methods (*Alzahrani & Hong, 2017*).

## LITERATURE REVIEW

This section presents and critically analyzes existing research studies to detect the attacks in the three categories of IDS methods mentioned above.

**Table 1 Summary of IDS techniques.**

| Methods | Signature-based | Anomaly-based | Hybrid-based |
|---|---|---|---|
| Advantage | ✓ Easy to implement in real-time<br>✓ Low cost | ✓ Effective against unknown attacks<br>✓ Effective against new attacks without database update | ✓ High accuracy rating |
| Limitation | ✓ Ineffective against zero-day attacks<br>✓ Must update the database for new attacks | ✓ Difficult to implement at run-time<br>✓ Detection accuracy affected by the number of collected features in dataset | ✓ Cost is high |

## Signature-based detection

*Tanrıverdi & Tekerek (2019)*, *Bakshi & Dujodwala (2010)*, and *Modi et al. (2013)* proposed a signature-based detection method. *Tanrıverdi & Tekerek (2019)* presented the detection of web attacks using a blockchain-based attack detection model. The signatures listed in this study are automatically updated by blockchain technology. An additional advantage of this proposed method is that it can be used against zero-day attacks. *Bakshi & Dujodwala (2010)* presented a method to distinguish between the normal and abnormal traffic in VMs. It uses Snort to analyze the collected traffic to determine the attack. The virtual server then drops packets coming from the specified IP address. *Modi et al. (2013)* presented a method to detect known attacks and derivatives of known attacks. It also uses a Snort tool to detect the known attacks from network traffic. The detected attack is input to a signature DB to predict derivatives of the attack by using signature as a priority.

## Anomaly-based detection

*Hong et al. (2017)* and *Kemp, Calvert & Khoshgoftaar (2018)* presented an anomaly-based solution to detect slow HTTP attacks, a type of DDoS attack. *Hong et al. (2017)* developed a software-defined networking (SDN) controller; however, *Kemp, Calvert & Khoshgoftaar (2018)* deployed the proposed model using machine learning techniques. It selected eight classification algorithms for predictive models: random forest, decision trees, K-nearest neighbor, multilayer perceptron, RIPPER (JRip), support vector machines, and Naïve Bayes. The authors used the Weka machine learning toolkit to build these models. ANOVA was used to compare the values of slow attack detection among the eight models. They evaluated the models by area under the receiver operating characteristic curve (AUC), receiver operating characteristic (ROC) curve graphs, true positive rate (TPR), and false positive rate (FPR).

*Singh, Jeong & Park (2016)*, *Lima Filho et al. (2019)*, *Wang et al. (2014)*, and *Sreeram & Vuppala (2019)* presented anomaly-based solutions to detect HTTP attacks. *Singh, Jeong & Park (2016)* used a multilayer perceptron with a genetic algorithm (MLP-GA)-based method for detecting DDoS attacks on incoming traffic. Authors in *Singh, Jeong & Park (2016)* identified four features to detect application-layer attacks; first is the number of HTTP counts, referring to the count number of requests per IP address. It is assumed that any single IP address that sends more than 15–20 HTTP GET/POST requests is an attack. Second is the number of IP addresses, referring to the number of IP addresses in small

windows time. It is assumed the attacks have more than 20 IPs in windows time. The third was the constant mapping function; the attacker's ports are different from legitimate users' as the one used by the attacker is varied and remains open. The fourth is fixed frame length; codes with fixed frame length are considered as an attack.

In comparison, *Lima Filho et al. (2019)* proposed an online smart detection system for DoS/DDoS attack detection. The detection approach used the random forest tree algorithm to classify various types of DoS/DDoS attacks such as flood TCP, flood UDP, flood HTTP, and slow HTTP. However, *Wang et al. (2014)* proposed a detection scheme for HTTP-flooding (HTTP-Soldier) based on web browsing clicks. HTTP-soldier used the large-deviation principle of webpage popularity to be able to distinguish between normal and abnormal traffic. The large-deviation probability-based detection may affect some normal users. The authors mentioned that their proposed scheme could not detect a single uniform resource locator (URL) attack. The false positive of a Multi-URL attack with the most popular webpages is 12.2%, but the false positive of Multi-URL attack with the least popular webpages is at 17.1%. This solution can achieve high performance in Multi-URL attacks with the most popular web pages. *Sreeram & Vuppala (2019)* used bio-inspired machine learning metrics to detect HTTP flood attacks to achieve fast and early detection. Authors in *Sreeram & Vuppala (2019)* adopted the Bat algorithm, which has low process complexity, as a bio-inspired approach.

*Choi et al. (2014)*, *Aborujilah & Musa (2017)*, and *Sahi et al. (2017)* presented a cloud-based flood attack detection method. *Choi et al. (2014)* proposed a method to integrate the detection of DDoS flood attacks and MapReduce processing in a cloud computing environment. The proposed framework consists of three parts: first is the packet and log collection module (PLCM), which analyses packet transmission and web server logs in the first part. Second is the pattern analysis module (PAM), which produces the pattern for DDoS attack detection. Finally is the detection module (DM), which detects DDoS attacks by comparing them with a normal behavior model. However, *Aborujilah & Musa (2017)* presented the detection based on the covariance matrix approach. The proposed detection was divided into training and testing phases. A training phase aimed to construct a normal network traffic profile. The testing phase was to detect any abnormal traffic by the deviation between the normal and any other network traffic. The normal traffic is captured from end-users browsing the Internet in their cloud, whereas the flooding attack traffic is generated using the PageRebooter tool. It was evaluated by using the confusion matrix and present results for an internal and external cloud environment, while *Sahi et al. (2017)* proposed a detection model for TCP flood DDoS attacks. This model selected different classifiers, the least squares support vector machine (LS-SVM), Naïve Bayes, K-nearest, and multilayer perceptron.

*Lin, Ye & Xu (2019)*, *Li et al. (2019)*, and *Nawir et al. (2019)* presented an anomaly-based detection method for detecting DDoS attacks. The proposal of *Lin, Ye & Xu (2019)* and *Li et al. (2019)* used deep learning techniques. *Lin, Ye & Xu (2019)* used long short-term memory (LSTM) to build the neural network model, which is a specific recurrent neural network structure (RNN). *Li et al. (2019)* used LSTM and gated recurrent units (GRU) recurrent neural networks. It used BGP and NSL-KDD datasets. The best

accuracy achieved was using the BGP dataset in the range of 90–95%. However, the proposed of *Nawir et al. (2019)* used machine learning algorithms. The authors in [Reff9+] selected five machine learning algorithms to include Naïve Bayes (NB), averaged one dependence estimator (AODE), radial basis function network (RBFN), multi-layer perceptron (MLP), and J48 trees. A comparison was drawn between these algorithms based on accuracy and processing time. A UNSW-NB15 dataset was selected in this experiment.

*Haider et al. (2020)* proposed a deep convolutional neural network (CNN) framework for efficient DDoS attack detection in SDN. This proposed framework has been evaluated using hybrid state-of-the-art algorithms on CICIDS2017 dataset. *Hwang et al. (2020)* proposed an unsupervised deep learning model for early network traffic anomaly detection, namely D-PACK based on CNN. The experimental results show low false-positive rate and high accuracy. *Novaes et al. (2020)* proposed using short-term memory and fuzzy logic for DDoS attack detection and mitigation in SDN. The proposed system consists of three phases: Characterization, anomaly detection, and mitigation. The evaluation of this system has been conducted using CICDDoS2019 dataset with archived accuracy of 96.22%.

## Hybrid-based detection

*Hatef et al. (2018)*, *Zekri et al. (2017)*, and *Novaes et al. (2020)* proposed a hybrid-based detection system. While *Hatef et al. (2018)* and *Zekri et al. (2017)* deployed cloud environment approaches. *Hatef et al. (2018)* is a hybrid intrusion detection approach in cloud computing (HIDCC). The applied detection was a combination of signature-based and anomaly-based detection techniques. The Snort tool is used for known attacks (signature-based detection) by employing the Apriori algorithm to generate a pattern from derived attacks. Both clustering and classification algorithms are applied for the undetectable attack through Snort. The clustering module receives and determines the input packet based on the sample vector. Then the classifier module determines the final class of the packet through algorithm C4.5 as a decision tree classifier according to the found cluster. However, *Zekri et al. (2017)* proposed using machine learning for anomaly detection and Snort technique for signature detection. Three algorithms were selected: decision tree, Naïve Bayes, and K-means algorithms. The decision tree achieved the best accuracy. The proposed *Saleh, Talaat & Labib (2019)* was applied in real time and deployed in three stages. First, the Naïve Bayes feature selection (NBFS) technique was employed to reduce the dimensionality of sample data. Second, optimized support vector machines (OSVM) were used to reject the noisy input sample as it might have caused misclassification. Finally, the attacks were detected by prioritized K-nearest neighbors (PKNN) classifier. This proposed scheme takes time in the first and second stages at feature selection and outlier rejection before attack detection.

This study will explore the use of data mining techniques (classification) in real-time detection. We will focus on the network/transport level as it is the core layer of network architecture. There are few existing studies on volume-based network/transport-level DDoS attack detection in the cloud environment. Moreover, there are very few studies that have proposed online detection with a high detection rate. This study will employ different

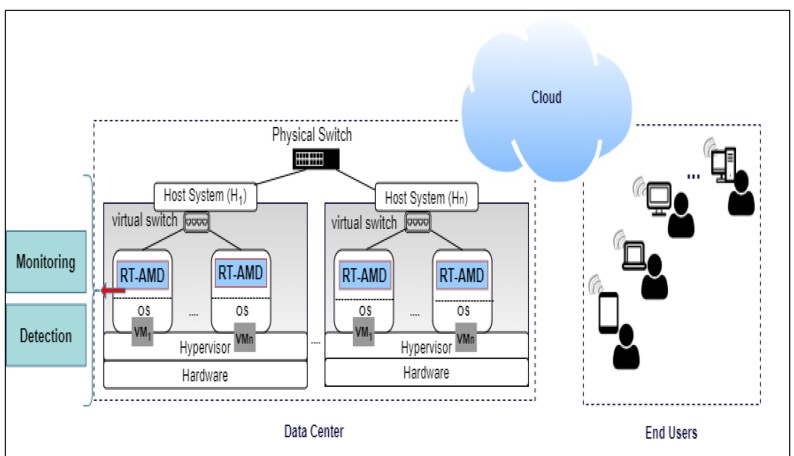

**Figure 2 Overview of RT-AMD framework.**

classification algorithms (machine learning) that suit our needs to build detection models. We will then evaluate these models by comparing them against two main factors: efficiency of detection and detection rate.

# PROPOSED FRAMEWORK

This section explains the RT-AMD model framework by presenting its components and the employed data mining classification methods.

## Main components

The proposed RT-AMD model consists of two main components: monitoring and detection.

1. The monitoring component is responsible for monitoring the network traffic for requests coming to the webserver on the cloud and extracting the traffic from the network log. If the traffic from the log is found in the Blacklist, an alert with corresponding information is sent to the cloud admin. If it is not in the Blacklist, it will move into the next component, detection.

2. The detection component uses trained classifiers to detect if the incoming traffic behavior is normal or abnormal. When abnormal behavior occurs, it will alert the system, send information to the admin, then update the Blacklist with the new traffic information. Otherwise, it can access the cloud and benefit from its services. An overview of the proposed environment is shown in Fig. 2.

At the beginning, the admin needs to log in/register to benefit from the RT-AMD tool. Once the registration is done, RT-AMD will send a verification code for the entered email to ensure that the email address is correct. The tool will then start monitoring the network traffic to detect any malicious behavior. Once an attack occurs, the RT-AMD will detect it, then send all information assigned to this traffic to the admin email. This makes it easier for the admin to act on any malicious behavior. A flowchart of the proposed detection framework is shown in Fig. 3.
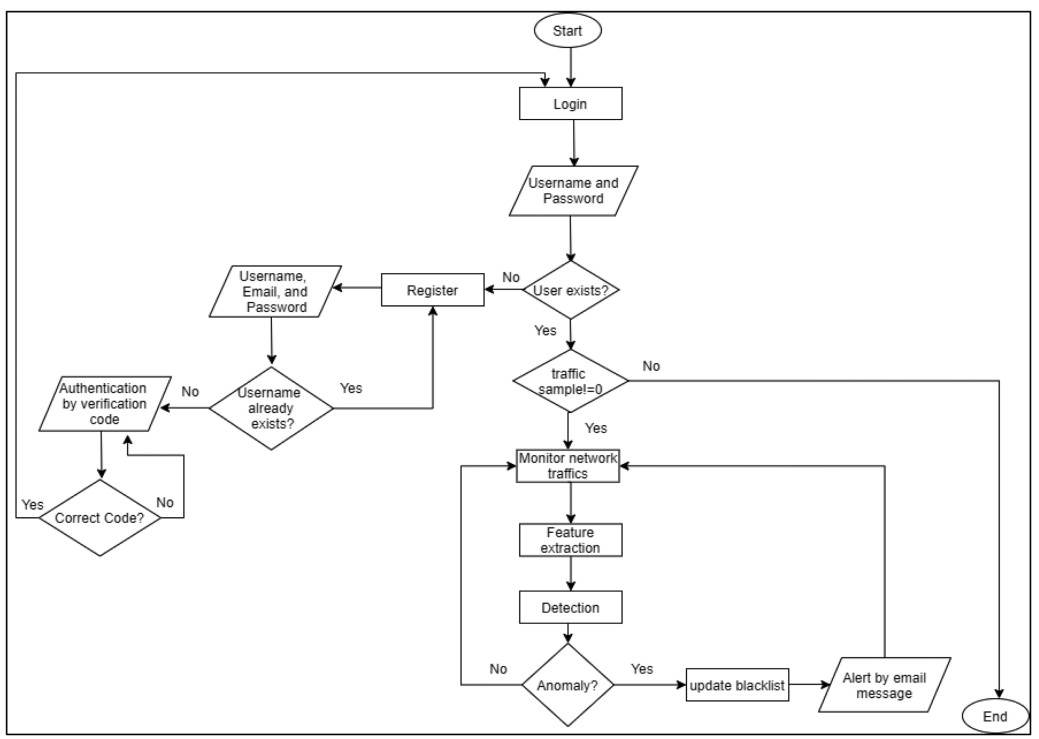

**Figure 3 Flowchart of RT-AMD framework.**

## Data mining classification methods

Data mining analyzes large volumes of data to identify and predict any threats; this helps to solve problems and reduce risks. Data mining can answer questions that have typically been time-consuming to address manually by using several statistical techniques to analyze data in various ways.

Recently, high arrival rates of online data streams have imposed high resource requirements on data mining processing systems. Datastream mining (also known as stream learning) is a technique of extracting knowledge structures from an unbounded and ordered sequence of data that exists over time (stream data) (*Gomes et al., 2017*). Some differences between stream data mining and traditional data mining are shown in the following (*Ramírez-Gallego et al., 2017*):

1. Machine learning of streaming scenarios cannot retrieve all of the data of the dataset in advance. Data chunks are available in a stream, one by one, or bundle by block.

2. Data arriving over time in streams may be unlimited in their number, resulting in difficulty storing all arrival data in the memory.

3. Data from streams need to be analyzed quickly to provide real-time response and prevent data waiting.

4. Some incoming stream data may lack accurate class labels because of the label query's high cost for each data stream.

Building knowledge from stream data mining is called incremental learning (*Ramírez-Gallego et al., 2017*). Incremental learning has received much attention from both academia and industry. It is a machine learning approach where knowledge is applied as new instances arrive, and what has been learned is updated according to the new instances (*Xie & Lam, 2006*).

There are several techniques in data mining, such as regression, clustering association, and classification. Different techniques serve different purposes. However, most data mining techniques usually applied for this area are classification techniques. Classification is a type of supervised learning that predicts the class label to which data belongs. The classifier works by obtaining a training dataset containing several attributes and class labels. The classifier then tests the dataset to evaluate the model.

The proposed framework will employ the classifiers in the detection component of the framework with two class labels. The attributes assigned to normal behavior are labelled "normal" and those to abnormal behavior "anomaly." Some classifiers were found to have better detection results than others based on the recommendations in the related works (*Kemp, Calvert & Khoshgoftaar, 2018*; *Lima Filho et al., 2019*; *Saleh, Talaat & Labib, 2019*). We selected the following: Naïve Bayes, decision trees, K-nearest neighbor, and random forest. These classifiers have been selected to build the predictive models.

## DATASET COLLECTION

A DDoS-2020 dataset is a network/transport-level dataset that authors have assembled and that contains of two types of traffic: attack and normal, with 70,020 records of traffic. Attack traffic consists of 27,169 instances, and the normal traffic consists of 42,851 instances. We have collected attack traffic from the CAIDA DDoS Attack 2007 dataset and collected normal traffic by capturing packets using Wireshark. The Center for Applied Internet Data Analysis Dataset "DDoS Attack 2007" (*CAIDA: Center for Applied Internet Data Analysis, in press*) contains an approximately one-hour collection of anonymized (abnormal) traffic from a DDoS attack on August 4, 2007 (*CAIDA: Center for Applied Internet Data Analysis, in press*). Wireshark is one of the most popular open-source network analyzer tools under the GNU General Public License (GPL) (*Munz & Carle, 2008*). Wireshark captures packets using the "pcap" library and different network media types, including Ethernet, Wi-Fi, Bluetooth, and others (*Wireshark, in press*).

The DDoS-2020 dataset contains information corresponding to each packet: source IP address, destination IP address, protocol type (ICMP, TCP, and DNS), packet length, packet timestamp, and label to determine whether the traffic is normal ("0") or attack ("1"). We have 29,554 instances of TCP, 20,727 instances of ICMP, and 19,739 instances of DNS with 0% missing value. Figure 4 shows the distribution of TCP, ICMP, and DNS protocols in the dataset.

The dataset contains two types of traffic: attack traffic consists of 27,169 instances and normal traffic consists of 42,851 instances. The label for normal traffic is = 0 and the label for attack traffic is = 1. The protocol's distribution ratio on normal traffic and the dataset's attack traffic are shown below in Fig. 5.
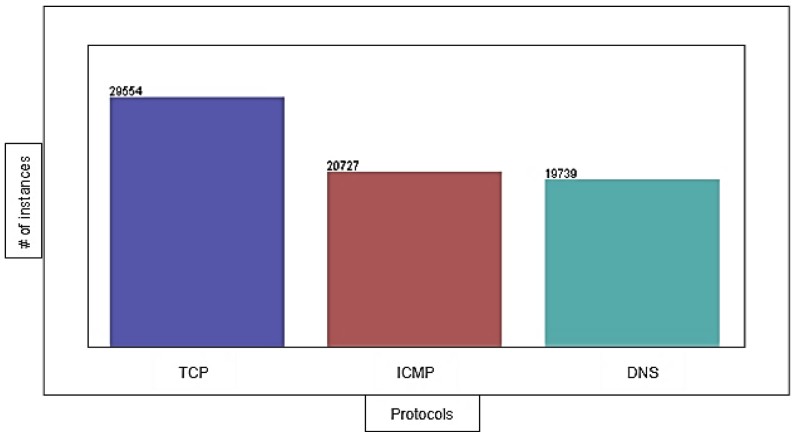

**Figure 4 The distribution of protocols in the dataset.**

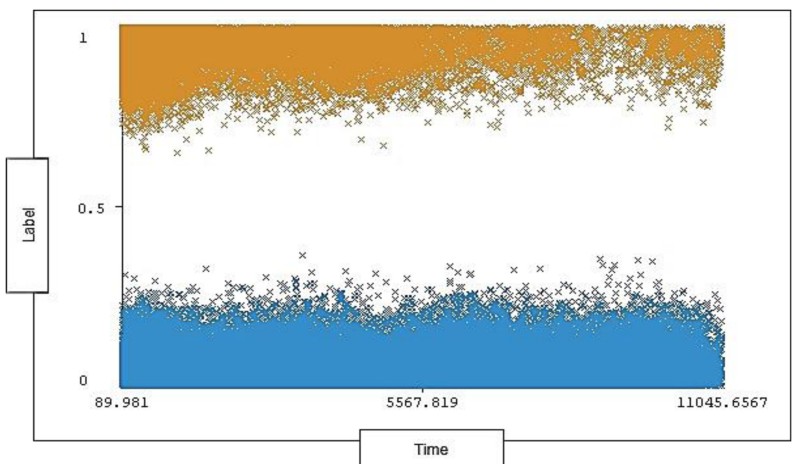

**Figure 5 The protocol's distribution ratio on normal and attack traffic.**

The timestamp range in the dataset is between 89.981 and 11,045.6567 seconds and 0% missing value. Each timestamp has a range of instances for each type of traffic (attack, normal). The timestamp distribution ratio on normal traffic and attack traffic in the dataset is shown below in Fig. 6.

The range of length is between 46 and 1,800 with 1,190 distinct and 0% missing value. Each length ranges across a group of instances for each type of traffic (attack, normal). The length distribution ratio on normal traffic and the attack traffic in the dataset is shown below in Fig. 7.

During the collection and cleaning of the DDoS-2020 dataset, we were determined to distribute the elements' ratio in a balanced manner. The balance is considered in the distribution of traffic between instances (normal and attack) and the distribution of individual protocol that has range of instances for attack and normal traffic. Figure 8 shows the distribution ratio of attacks and normal traffic for each TCP, ICMP, and DNS protocol.

Figure 9 shows the distribution ratio of attacks and normal traffic for each timestamp.

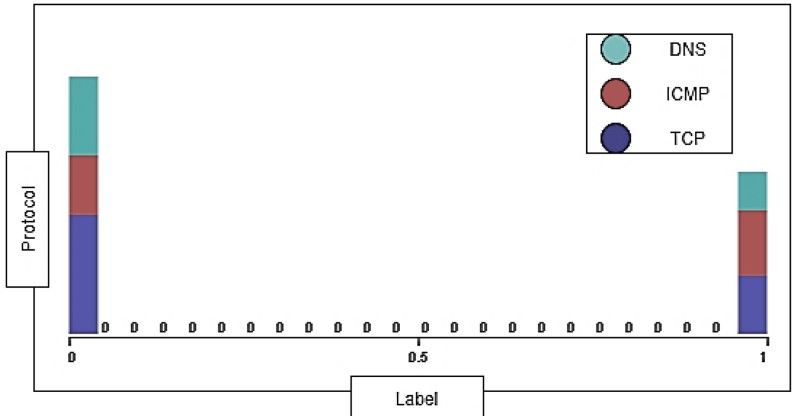

**Figure 6 Distribution of timestamp over labels.**

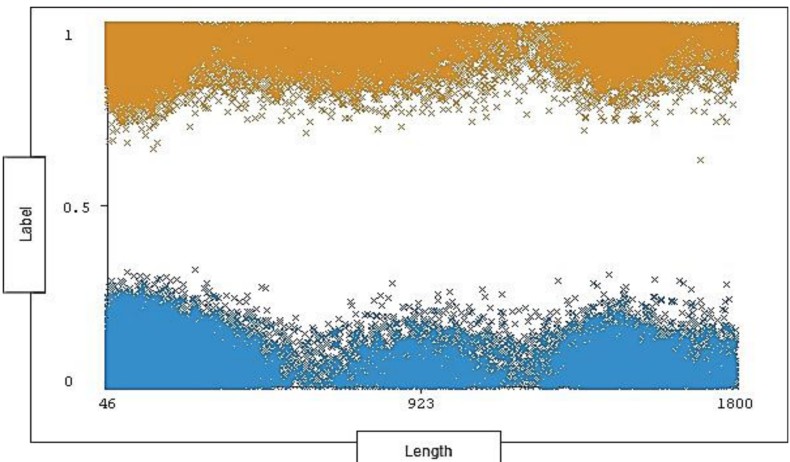

**Figure 7 Distribution of length over labels.**

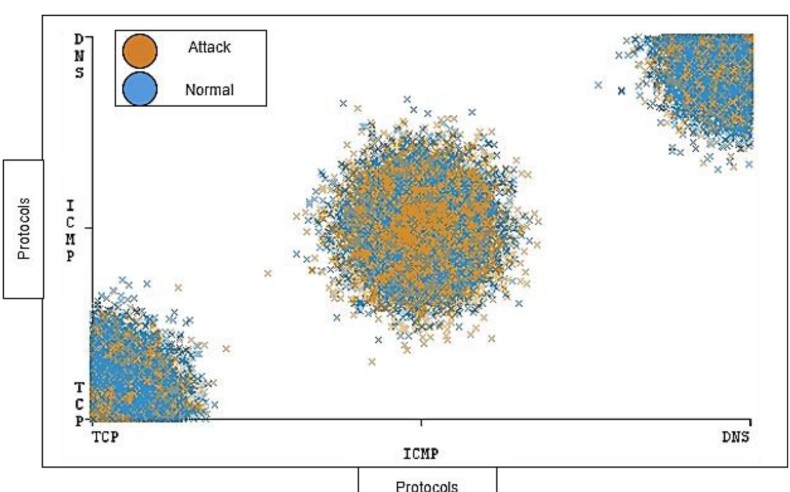

**Figure 8 Traffic distribution of protocols.**

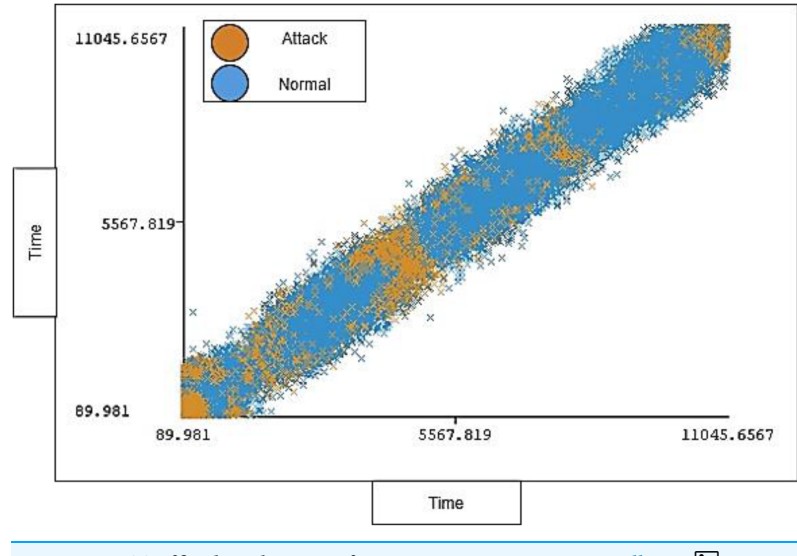

**Figure 9 Traffic distribution of timestamp.**

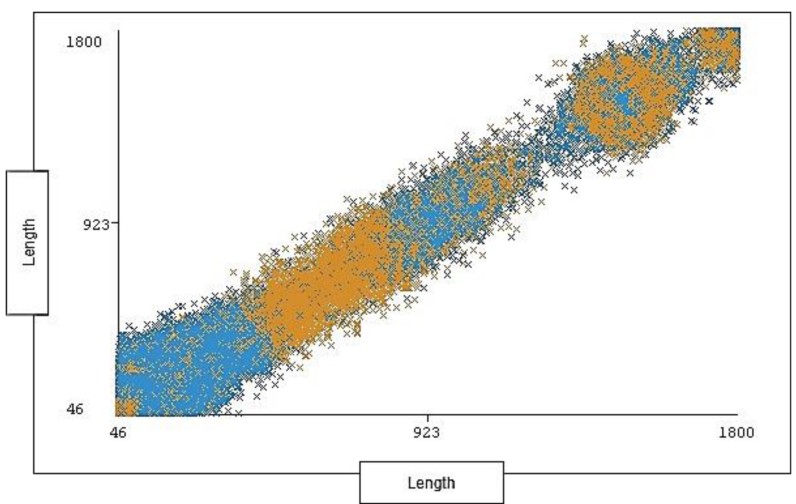

**Figure 10 Traffic distribution of length.**

Figure 10 shows the distribution ratio of attacks and normal traffic for each length range.

The timestamp and the length of traffic is also notable; the distribution ratio of the timestamp and length for each protocol should be semi-balanced. Figures 11 and 12 show the distribution ratio of timestamp and length over protocols sequentially.

## EVALUATION RESULTS

RT-AMD model is implemented using the Python programing language, SQLite, and GCP. Python is a powerful language; it contains many libraries for machine learning. One of the libraries that is well-known for real-time learning purposes is Scikit multi-flow. The evaluation of this model was in the GCP environment.
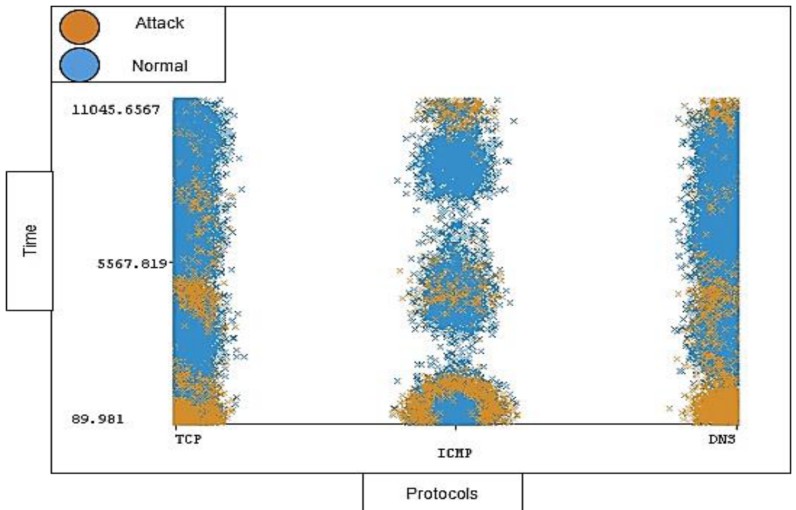

**Figure 11 Distribution of timestamp over protocols.**

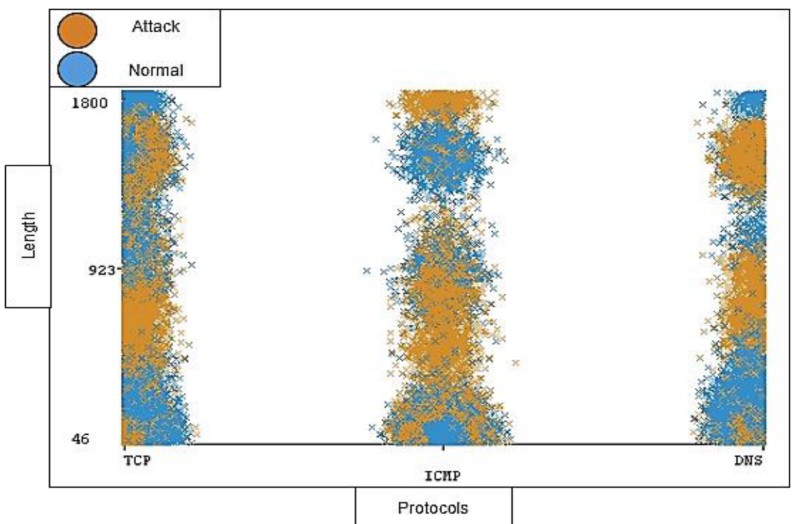

**Figure 12 Distribution of length over protocols.**

As mentioned above, the RT-AMD tool was evaluated by the selected machine learning algorithms. Naïve Bayes, decision tree, K-neighbors, and random forest were selected for this evaluation. We experimented with the RT-AMD tool in three situations: offline localhost, online localhost, and online remote virtual host created by GCP. Configuration of localhost was with Microsoft Windows 10 Pro operating system, 24.0 GB RAM, and Intel(R) Core(TM) i7-7500 processor, and the remote virtual host was configured with e2-medium machine type, 2 vCPUs, and 4 GB memory.

The evaluation measured accuracy and performance. The random forest achieved the best accuracy in incremental learning either on localhost or remote virtual host at around 99.38%. However, K-neighbors achieved the best accuracy in offline learning. Table 2 and Fig. 13 show the details for each experiment. The Naïve Bayes achieved the efficient

**Table 2 RT-AMD accuracy for each experiment.**

| Algorithms | Local offline testing accuracy (%) | Local online testing accuracy (%) | Online cloud testing accuracy (%) |
|---|---|---|---|
| Naïve bayes | 77.20 | 81.88 | 81.86 |
| Decision tree | 92.99 | 97.90 | 97.89 |
| K-neighbors | 93.87 | 98.48 | 98.49 |
| Random forest | 92.37 | 99.39 | 99.38 |

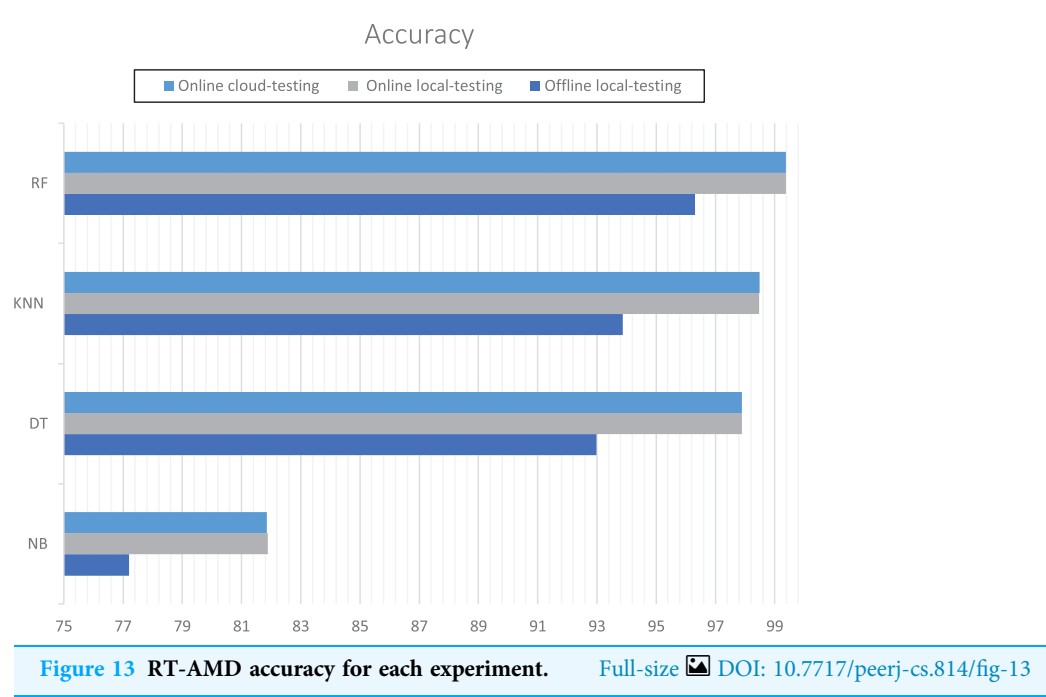

**Figure 13 RT-AMD accuracy for each experiment.**

**Table 3 Execution time details.**

| Algorithms | Local online testing execution time | Online cloud testing execution time |
|---|---|---|
| Naïve bayes | 22.32 s | 12.08 s |
| Decision tree | 27.28 s | 14.81 s |
| K-neighbors | 297.94 s | 260.84 s |
| Random forest | 567.64 s | 741.84 s |

execution-time for online learning: 12.08 s for cloud testing and 22.32 s for local testing. Table 3 shows the details of execution time for local online testing and online cloud testing.

The proposed tool is tested with different datasets; our DDoS-2020 dataset and NSL-KDD dataset (*University of New Brunswick, in press*). The NSL-KDD dataset contains 125,964 samples, among which 67,343 are normal and 58,621 attacks. The NSL-KDD is a useful dataset and popularly used in previous studies (*Haider et al., 2020*; *Saleh, Talaat & Labib, 2019*). It is a new version of the KDD'99 dataset. Table 4 and Fig. 14 show the details of accuracy for each dataset in cloud testing.

**Table 4 Accuracy and execution time for each dataset in cloud testing.**

| Algorithms | DDoS-2020_Dataset accuracy (%) | NSL-KDD_Dataset accuracy (%) |
|---|---|---|
| Naïve bayes | 81.86 | 89.42 |
| Decision tree | 97.89 | 97.25 |
| K-neighbors | 98.49 | 95.23 |
| Random forest | 99.38 | 99.30 |

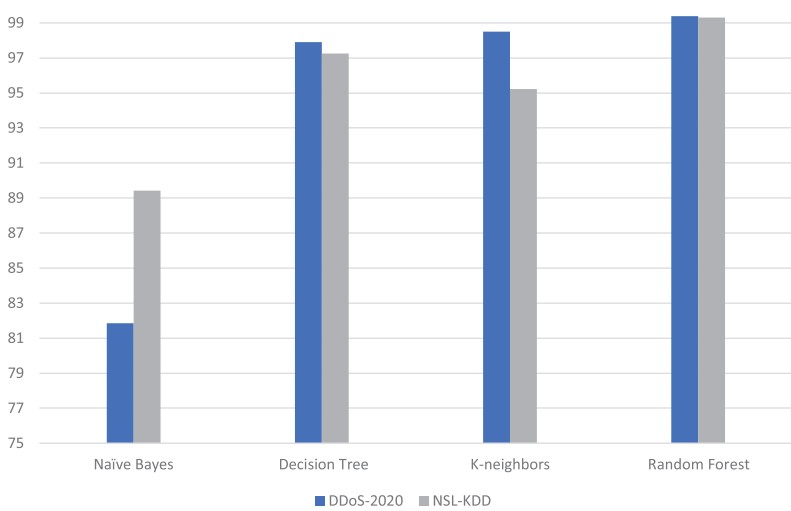

**Figure 14 Accuracy and execution time for each dataset in cloud testing.**

## DISCUSSION

As we mentioned above, the experimental RT-AMD tool was used in three different situations; offline localhost, online localhost, and online remote virtual host created by GCP. We achieved 99.38% accuracy with real-time detection in a cloud environment for the random forest. The accuracy for online detection is much higher than offline detection; this is because of Scikit-multi-flow library incremental learning characteristics. The execution time of random forest on the cloud was worst for many reasons, such as the virtual machine's abilities and the way random forest algorithms work.

Our model achieved the best accuracy with a real-time response on the cloud environment in comparison with related work. This is due to the model work and the machine learning algorithm's efficiency using the Scikit-multi-flow library. We have seen above the same model results offline using a Scikit-learn and online using the Scikit-multi-flow library. The latter library features incremental learning that gradually improves the algorithms' performance with runtime, thus improving results. Table 5 and Fig. 15 show the details of this comparison.

**Table 5 Comparison of the result with related work.**

| Related work | Detection method | Dataset | Accuracy (%) | Cloud-based | Online |
|---|---|---|---|---|---|
| *Singh, Jeong & Park (2016)* | Anomaly-based detection using random forest tree algorithm | Customized dataset | 96.5 | × | ✓ |
| *Lima Filho et al. (2019)* | Anomaly-based detection using genetic algorithm | CAIDA2007 | 98.04 | × | × |
| *Wang et al. (2014)* | Anomaly-based detection based on web browsing clicks | Used the weblog of CDU website (www.cdu.edu.cn) as a simulation dataset and replayed the dataset with NS-34 | 94.9 | × | × |
| *Sreeram & Vuppala (2019)* | Anomaly-based detection using bio-inspired | CAIDA 2007 | 94.8 | × | × |
| *Sahi et al. (2017)* | Anomaly-based detection using four different classifiers LS-SVM, Naïve Bayes, K-nearest, and multilayer perceptron | N/A | 97 | ✓ | × |
| *Lin, Ye & Xu (2019)* | Anomaly-based detection used deep learning techniques LSTM | CSE-CIC-IDS2018 | 96.2 | × | × |
| *Li et al. (2019)* | Anomaly-based detection used deep learning techniques LSTM and GRU | BGP | 95.21 | × | × |
| *Nawir et al. (2019)* | Anomaly-based detection using machine learning algorithms including Naïve Bayes, averaged one dependence estimator, radial basis function network, multi-layer perceptron, and J48 trees | UNSW-NB15 | 97.26 | × | × |
| *Hwang et al. (2020)* | Hybrid-based detection using machine learning for anomaly detection and Snort technique for signature detection | N/A | 98.8 | ✓ | × |
| *Novaes et al. (2020)* | Hybrid-based detection using K-nearest neighbors classifier for detection | NSL-KDD | 95.77 | ✓ | ✓ |
| RT-AMD (our proposed) | Anomaly-based detection using machine learning algorithms including Naïve Bayes, decision tree, K-neighbors, and random forest | DDoS-2020 | 99.38 | ✓ | ✓ |

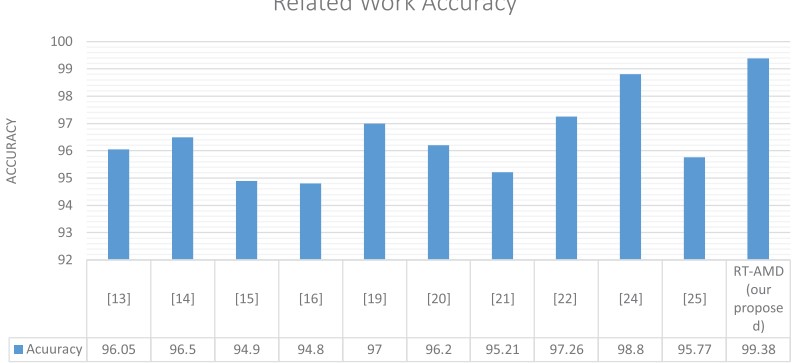

**Figure 15 Comparison of the result with related work.**

## CONCLUSION AND FUTURE WORK

This study discusses the issues surrounding DDoS attacks on cloud environments, presenting the main types of DDoS attacks and the challenges and risks faced. Further, it

reviews some of the previous techniques detecting DDoS attacks. Machine learning is one of the most common techniques used to detect DDoS attacks. Furthermore, incremental learning is one of the best strategies to learn and classify in real-time. This study's main contributions are to evaluate machine learning algorithms for the dataset collected and investigate the results with related works. Furthermore, we improve outcomes and reach real-time attack detection by using incremental learning.

The RT-AMD model is proposed to detect DDoS attacks on the cloud environment using machine learning techniques. Four machine learning algorithms were selected to evaluate this model: Naïve Bayes, decision tree, K-neighbors, and random forest. The RT-AMD model was developed by python, SQLite databases, and GCP to detect and alert of the DDoS attacks and test on the cloud environment platform. It was evaluated by using two datasets, the DDoS-2020 and NSL-KDD dataset. The DDoS-2020 dataset has been collected with two ranges of traffic (attack and normal), with three distinct network/transport protocols: TCP, ICMP, and DNS. The attack traffic were obtained from CAIDA DDoS Attack 2007 and the normal traffic were obtained by using Wireshark.

As a result, the RT-AMD model achieved high accuracy in DDoS-2020 dataset testing and NSL-KDD dataset testing. The random forest algorithm obtained the best accuracy, reaching 99.38% with the DDoS-2020 dataset and 99.30% with the NSL-KDD dataset. This model achieved real-time detection without the negative effect on accuracy by using an incremental learning strategy, and without needing pre-training machine learning.

There are various ways to extend the study presented in this research. These include extending dataset samples to include different types of DDoS, and evaluating and testing this model on other cloud computing-related environments such as mobile cloud computing (MCC), a combination between cloud computing and mobile computing.

### Funding
This work was supported by the Institutional Fund Projects under grant number (IFRPC-114-612-2020). Technical and financial support was received from the Ministry of Education and King Abdulaziz University, Jeddah, Saudi Arabia. The funders had no role in study design, data collection and analysis, decision to publish, or preparation of the manuscript.

### Grant Disclosures
The following grant information was disclosed by the authors:
Institutional Fund Projects: IFRPC-114-612-2020.
Ministry of Education and King Abdulaziz University, Jeddah, Saudi Arabia.

### Competing Interests
The authors declare that they have no competing interests.

## Author Contributions

- Omaimah Bamasag conceived and designed the experiments, performed the experiments, prepared figures and/or tables, and approved the final draft.
- Alaa Alsaeedi conceived and designed the experiments, performed the experiments, performed the computation work, prepared figures and/or tables, and approved the final draft.
- Asmaa Munshi conceived and designed the experiments, performed the experiments, prepared figures and/or tables, and approved the final draft.
- Daniyal Alghazzawi analyzed the data, authored or reviewed drafts of the paper, and approved the final draft.
- Suhair Alshehri analyzed the data, authored or reviewed drafts of the paper, and approved the final draft.
- Arwa Jamjoom analyzed the data, authored or reviewed drafts of the paper, and approved the final draft.

## Data Availability

Raw data and code are available in the Supplemental Files.

## Supplemental Information

Supplemental information for this article can be found online at http://dx.doi.org/10.7717/peerj-cs.814#supplemental-information.

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
