# Peer review of "Real-time DDoS flood attack monitoring and detection (RT-AMD) model for cloud computing"

_PeerJ Computer Science, doi:10.7717/peerj-cs.814_

## Round 0.1 · original submission · Minor Revisions

The reviewers have found the article very interesting and ready for publication subject to minor corrections.

Reviewer 1 ·

Basic reporting

The analysis of the findings is very interesting. Many challenges of this proposed method are covered. Literature reference is relevant, but does not cover all the topics, such as AI, and is not so up to date.

Experimental design

As a concept, it is well designed. I found the topic interesting and definitely an area of growth.

Validity of the findings

Despite the research effort and the achieved high rate of accuracy, there are a number of challenges that need to be addressed, such as execution-time for online learning of random forest algorithm, the lack of up-to-date real-world datasets for training, what is the rate of false-positive and false-negative, there are scalability issues or performance?

Reviewer 2 ·

Basic reporting

The paper is well structured and easy to read. Also, the use of English is quite good. The introduction provides a great, generalized background of the topic and the motivations for this study are clear. A minor comment concerns the References part, where it would have been better to have used more recent literature with more up-to-date knowledge.
The experimental part is well written and is appropriate for the study.
To conclude, this research work apparently fulfills the purpose for which it was carried out. It would be interesting to see in the future, the use of more machine learning classifiers in building the Real-Time DDoS flood Attack Monitoring and Detection RT-AMD predicting model or even use more datasets for executing the same experiments.
For the above reasons I strongly recommend the acceptance of this paper.

Experimental design

The research question of this research is well defined and undoubtedly adds knowledge to an otherwise quite explored scientific area, that of attack monitoring and
detection in cloud computing. The methods used, although not innovative, have certainly been used in the right way and the overall investigation is performed to a high technical standard.

Validity of the findings

Throughout the research there is no replication of pre-existing knowledge.
Conclusions are both well stated and linked to original research question.

---

## Round 0.2 · accepted · Accept

Your article is now ready for publication